# Process Planning in Industry 4.0—Current State, Potential and Management of Transformation

**Maja Trstenjak [1,\*]**, **Tihomir Opetuk, Hrvoje Cajner and Natasa Tosanovic**

Faculty of Mechanical Engineering and Naval Architecture, University of Zagreb, Ivana Lucica 5,
10 000 Zagreb, Croatia; tihomir.opetuk@fsb.hr (T.O.); hrvoje.cajner@fsb.hr (H.C.); natasa.tosanovic@fsb.hr (N.T.)
* Correspondence: maja.trstenjak@fsb.hr; Tel.: +385-98-908-2564

**Abstract:** The implementation of the Industry 4.0 concept enables the flexibility, modularity and self-optimization of the manufacturing process. Process planning, placed in the value chain between construction and physical manufacturing, therefore, also demands digital transformation, while management of the transformation towards the new digital framework represents one of the most demanding challenges. Continuing the research on its structure and role within the smart factory, the main motivation for this work was to recognize the potential of the digital transformation of process planning elements, and to define the key dimensions that are essential for the readiness factor calculation and later transformational strategy formation, but also to recognize the current level of awareness of the Industry 4.0 concept among the process planners, along with the current use of its elements and key priorities for the transformation. The research has therefore been conducted in 34 Croatian metal machining companies, within which the influence of company size, level of education and familiarity with Industry 4.0 on final results and the stage of development have been investigated. The results have shown that the company size has a significant influence on the development stage and the use of certain elements wherein small and medium enterprises (SMEs) have already implemented certain digital elements, while they also tend to have a better fundamental infrastructure when using complex process planning methods, unlike others, which are still highly traditional. Organization and human resources have been ranked with the highest priority for change, while target goals for hardware and software have been set, with the managerial challenges of transformation defined and discussed.

**Keywords:** process planning; Industry 4.0; smart factory; readiness factor

## 1. Introduction

Industry 4.0 demands more or less radical changes within the working environment, depending on the current development level. Every company, whether manufacturing or service, must adapt in order to keep up with the market trends [1]. The achievement of the new concept is possible through a transitional digitization period, whose strategy must be optimized in order to avoid unnecessary time and money loss [2]. High investments might be challenging for small and medium enterprises (SMEs), but market leaders have already started with new digital concept implementation that will make them flexible, modular and give them the ability to predict future market trends, while remaining at the forefront and even widening the gap between them and the rest of the market [3]. The innovation management requires the recognition of fundamental drivers of earnings and sustainable growth, which will later have a direct impact on the increase of business effectiveness, also ensuring that it maintains its level of competitiveness within the market [4], while strategic management enables the successful implementation of the key elements, which can also be related to the implementation of Industry 4.0 in the working environment [5].

To develop an adequate strategic and investment plan, the screening of the current state of the company should be provided, which will later be compared to the ideal model of Industry 4.0 as a future target. Most of the research conducted so far [6] has been based on the company as a single system or part of the regional industrial benchmark system [7]. When considering the specific industrial fields, research has also been conducted for the logistics [8], ERP (enterprise resource planning) systems [9], manufacturing [10] and human resources departments [11]. Process planning, as a standalone department, has still not been included in this kind of research, although it has been discussed and analyzed as a part of the manufacturing process [12]. This is why the research conducted for the process planning as a single department is a novelty in the field, continuing from the ideal model of digital process planning previously presented in the literature [13].

Process planning in Industry 4.0 is an advanced CAPP (computer-aided process planning) system that collaborates with every part of the supply and value chain and is in direct relation to manufacturing planning and scheduling. This is why it is presented as "product planning", and enables the generation of an automatic process plan based on big data analytics that require the utilization of data stored within the product, tool, machine, knowledge and other databases. The predictive analytics assists the definition of the optimum process plan by the evaluation of real-time data about the local use of available machines and tools, as the time-to-market shortens with the production of novel smart products [14].

Advanced "product planning" systems of this kind demand changes within the company on the hardware, software and organizational levels. During the transitional period, the managerial activities are a key factor in optimal readiness factor calculation (pre-screening of the current state within the company), strategy definition and implementation. For the first phase of research, which enables the management to acknowledge how far from the new digital system companies currently are, research was conducted and the results are presented and discussed in this paper. The research was conducted within Croatian metal machining companies, but its structure and definition of key Industry 4.0 elements with their priorities can be useful as the basis for research in other regions or industry types, in order to create a wider regional benchmark.

This research aims to distinguish the current process planning methods and Industry 4.0 elements currently used in Croatian metal machining companies, in order to obtain a detailed overview of the current state of practice and examine the possibilities for digital concept strategy development and, finally, its implementation.

This paper is structured in three main thematic parts. In the first, a literature review of the current state of process planning in Industry 4.0 is given, as well as the use and importance of CAPP systems within the digital concept, alongside its challenges and potentials, in order to recognize and emphasize the scientific gap that motivated the research. The second section provides a description of the research methods used, the survey structure and content. The results of our research into the current state of process planning in Croatian metal machining companies are presented in terms of the influence of company size, familiarity with Industry 4.0 and examinees' educational level on the results tested. In the third section, the results are discussed, compared to international, previously conducted research and target values for key Industry 4.0 elements and also compared to the current state. The emphasis is placed on process planning and the investigation is conducted within a specific industrial sector, in order to obtain a detailed overview of the current state of practice and examine the possibilities for digital concept strategy development and, finally, its implementation.

## 2. Process Planning and Industry 4.0

The traditional approach to process planning relies on the knowledge, experience and intuition of a single person. A process planner deals with every activity individually, while in small companies, especially in the manufacturing craft, an owner is usually the person dealing with both process planning and scheduling activities. This particular person also generates the results (process and manufacturing plan), based only on their previous work experience within the very same company,

which may lead to an increase in waste and non-optimal process planning, due to the high level of subjectivity, the avoidance of new technologies and general mistrust of market trends [15].

This inflexible reliance on a single person for decision making can be one of the greatest barriers in the implementation of Industry 4.0, due the high level of resistance to change [16]. This is why the first step in the transitional process is to raise awareness of the future benefits of a new digital concept, in order to enable the strategic implementation of certain Industry 4.0 elements towards the final goal of the fully digital Industry 4.0 concept.

In the following sections, the development stages of process planning—which are crucial for a shift towards Industry 4.0—will be given, and their existence will later be examined in a specialized research, with results presented and discussed.

## 2.1. Computer-Based Process Planning Tools

The first step towards process planning digitalization is one of the most commonly used tools in the manufacturing industry. Use of CAD (computer-aided design) software enables the digital manipulation of technical drawings of products, and also provides easy accessibility to information about product design. It increases the productivity of both the designer and process planner and improves communication in this part of the value chain. The data from the CAD drawing can later be extracted and saved to corresponding databases, including data about the size, shape, quality attributes, constraints and other technological features of a product. The first CAD systems were oriented to two-dimensional drawings, while later developments were expanded to include three-dimensional model creation and manipulation. This step enables better visualization of a product, as well as the extraction of geometrical and technological features that can later be optimized in terms of improved and more effective planning, design and project management [17].

CAM (computer-aided manufacturing) systems, on the other hand, enable process simulation, machine tool control and operation assistance during planning, manufacturing, management, transportation and storage. Its basis is a three-dimensional model, which is processed with various types of data about machines, tools, fixtures, etc., to create an optimal process simulation which will later be generated manually as a process plan. Simulation also enables automatic machining time calculation, with control over the process from the raw material shape to the final product shape [18].

The integration of those two very important segments of process planning is realized in CAPP (computer-aided process planning) systems. The final result is the process plan (often also referred as process sheet/method sheet), which contains a listing of production operations, operation sequencing and associated machine tools with work regimes for a work part or assembly. This is an example of the process planning automatization which reduces preparatory time, while the advancement of its features leads to complete process planning in the Industry 4.0 system [19].

### CAPP and Industry 4.0

CAPP systems, which enable automatic process plan definition, are complex and their implementation—as with most Industry 4.0 features—requires very high investment, as well as an increased level of IT knowledge among the users and developers, compared to previous industrial demands. That is why there are only few evidences of its complete implementation and integration within manufacturing systems available so far, especially in SMEs and craft manufacturing.

Industry 4.0 demands implementation of its advanced version, with improved data collection, storage and processing in real time, from different parts of the value chain, in order to generate an optimal process plan. Therefore, there are certain challenges that need to be resolved prior to successful CAPP implementation within Industry 4.0 systems, which are recognized from the recently published scientific research.

There are three types of CAPP presented in the literature: variant, generative and hybrid. In the variant CAPP approach for a single product, the similarities with previously produced ones are identified, which implies that the new product can be produced using similar or even identical

manufacturing processes. The data is stored within different families of products that are recognized by special codes and definitions. In terms of Industry 4.0, this requires the development of large databases, as well as special feature recognition algorithms (classification methods) to enable the product family grouping and later appropriate process plan generation [20].

More advanced is the generative CAPP, which requires the use of algorithms, decision logic, formulas and geometry-based data to perform unique processing decisions, to take from the raw material to the finished product state. This requires no referral to previously produced products, while the single part specifications are mandatory input. These kinds of systems are very complex, while, in terms of the Industry 4.0, this approach also requires the development of large databases of rules and procedures with built-in knowledge. It is a system that synthesizes the process information in order to create a process plan for a new component automatically with very little human intervention. Decision logic and advanced decision support systems are key parts of the generative approach, which can later be extended, in terms of the Industry 4.0 digital twin, to enable accurate geometrical features recognition, automatic process plan definition and simulation of the machining in several iterations to enable the self-optimization of process plan [21].

Hybrid CAPP is the combination of variant and generative. In the first step, workpiece is associated to a family, which is described by the knowledge database that contains all possibilities for the manufacturing of the particular product [22].

Before the definition of process planning by the Industry 4.0 principle, several challenges in CAPP implementation are found in the literature. First and most important is the automatic recognition of geometrical features of product. Since no single widely-used commercial CAPP system has been developed, there are various approaches presented for the specific industrial needs. In the recent studies, Ma et al. have concluded that the traditional methods for building a CAPP system for turning are infeasible, due to the fixed feature library and hard-coded heuristic rules, while developing customizable approach for automatic process planning of rotational parts based on novel cell machinability analysis [23]. Zubair and Mansor noticed that the recognition of revolved regular-freeform surfaces is yet challenging, and therefore presented a new method that generates sub-delta volume using the volume decomposition method [24]. Al-Wswasi and Ivanov noticed the problem of recognition of features intersecting issues in automatic feature recognition (AFR) techniques. They presented a methodology that was written using C# coding, which provides geometrical and topological information of intermediate and final features from a smart interactive AFR (SI-AFR) system as an input [25]. Svrivastava and Komma focused on a problem of STEP (Standard for the Exchange of Product Data)-based exchange of product data, which carries high-level information about machining processes and features, and presented STEP AP238 as a more sustainable and reliable format than ISO14649. They have developed an interface for the generation of STEP AP238 code of conformance class-3 (a feature-based code) using different modules [26]. Shi et al. dealt with fact that the traditional feature recognition methods are computationally expensive and very complex, so they developed a novel method, a deep learning framework based on multiple sectional view (MSV) representation named MscNet, where 3D models are collected as an input and the information are combined via a neural network for recognition, which resulted in very small number of training samples, which outperformed the traditional recognition methods [27]. Pisec et al. presented a feature recognition method which, unlike the traditional methods based on the boundary representation based on the data from STEP file, is based on the rule-based algorithm. This generates the output of the preprocessor such as dimensions, coordinates and spatial position to the XYZ axis system [28]. Rampur and Reur presented a concept of automatic feature recognition using a CAD model neutral file in the format of ISO10303 STEP AP-214 to identify the role feature [29], Katataki and Abu Mansor used automatic generation of delta volume (DV) in the free form feature recognition [30], while Venu et al. developed the feature recognition system implemented in Java programming language for the B-spline surface features [31]. Behandish et al. focused on the hybrid manufacturing technologies (HM), which combine additive (AM) and subtractive (SM) manufacturing capabilities, where the

goal is to create the multimodal HM process plan represented by a finite Boolean expression of AM and SM manufacturing primitives, such that the expression evaluates to an "as-manufactured" artifact [32], while the feature recognition based on 3D convolution neural network has been developed by Zhang et al. [33].

This confirms that the development of optimal and extensively used CAPP system is still a work in progress, and that most of the main characteristics of Industry 4.0 such as big data manipulation, the internet of things and cloud computing can only enhance the faster development of the new automatized and digital process planning systems.

With CAPP systems as the core, several findings of process planning in Industry 4.0 have also been presented in the recent literature.

Lundgren et al. noticed that the lack of interoperability between different computer applications used in process planning and quality assurance results in information fragmentation, data duplication and potential data inconsistency. This represents one of the primary goals of Industry 4.0 implementation, where every segment of the value chain is connected; therefore, data are collected and analyzed in the real time to create an optimal process plan. They proposed a novel, model-driven approach for process planning, integrating quality assurance, which emphasizes the application of digital models to create, represent and use the information of products, processes and resources, which has reduced the amount of data and document duplication with increase of the direct value [34]. Chen and Sun found increased complexity of manufacturing industry demands challenging and therefore developed an intelligent computer-aided process planning (i-CAPP) system, based on measuring the performance of manufacturability and efficiency in CNC machining by using hybrid-two-stage optimization algorithms—the traveling salesman problem and the innovative Tabu Search [35]. Gernhardt et al. enhanced the importance of cloud-based repository and distributed architectures to make data and information accessible. They presented a CAPP system that is supported by semantic approaches for knowledge representation and management, which, they assert, is especially well-suited to be used by SMEs. The importance of cloud-based collaborative planning is resolved within their system by knowledge-based production planning (KPP) [36]. The use of building information modelling (BIM) and radio frequency identification devices (RFID) and related technologies, in coordination with material flow processes and modeling with the development of a special database system, was developed by Chen et al. [37], while the importance of creation of sustainable production planning and control system based on smart technologies such as the internet of things, machine learning and cloud computing was presented by Oluyisola et al. [38].

Many authors have recognized the importance of cloud-based systems and virtual product development where Ahmed et al. developed the smart virtual product development (SVPD) system, which comprises three main models (design, knowledge and management), manufacturing capability analysis and process planning and product inspection planning [39]; Rodriguez et al. presented a methodology that connects process planning with an intelligent decision support system with implemented fuzzy logic and machine learning techniques [40]; Subramaian et al. developed a prediction-optimization framework for site-wide process optimization [41]; Milosevic et al. developed a cloud-based process planning system, which integrates knowledge sources with a production process using heuristic knowledge for optimizing process plans and selecting the best solutions [42].

Based on the literature review, it is certain that the Industry 4.0 concept enhances the required development of CAPP, which is needed to create a sustainable, digital and automated process planning system. Since there are many theoretical approaches available without much evidence of implementation and practice, the research of the current state, potentials and needs of the management of transformation was conducted within the industry, which will be described in the following chapters.

## 3. Research Design, Materials and Methods

The aim of the research was to examine the current state and potentials of process planning within the framework of Industry 4.0 in Croatian industry. The survey was created based on the theoretical

framework found in the literature (explained in the Chapter 2) and the "Process planning in Industry 4.0" concept from the previous research by authors, presented in reference [11].

Target group in this research were Croatian companies specialized in metal machining, regarding the presence of process planning in their system. The examinees have been found via the Croatian Chamber of Commerce's Digital Business Database [43]. Within the National Classification of Activities database, the category chosen, as the target population for the research is:

*C Manufacturing*
*25 Manufacturing of fabricated metal products, except racks and fittings*
*256 Treatment and coating of metals; metal machining*
*2562 Metal machining*

Additionally, as the digitalization of process planning, as a single department, has not yet been subject of this kind of research in the literature, this approach can later be conducted and the results used to make comparisons with other countries from the region or EU.

A total of 349 companies were listed in the database, out of which 243 were marked as working without active financial lockdown. The ones dealing with the trade of metal parts have been excluded from the research, as well as those whose contact data could not be found either in the database or online. This has limited the research on total of 119 companies–micro, small and large manufacturers. Process planning for the metal machining is also widely present in craft manufacturing in Croatia. This is why those subjects have also been included in the research, with 42 subjects having been found in this category.

Croatian law [44,45], based on directives from the European Union [46], defines enterprises in categories by size.

Micro enterprises have less than 10 employees and have an annual turnover of the amount of the equivalent of up to EUR 2,000,000, and/or have total assets/fixed assets in the amount of the equivalent of up to EUR 2,000,000.

Small enterprises have less than 50 employees and have an annual turnover of the amount of the equivalent of up to EUR 10,000,000, and/or have total assets/fixed assets in the amount of the equivalent of up to EUR 10,000,000.

Medium enterprises have less than 250 employees and have an annual turnover of the amount of the equivalent of up to EUR 50,000,000, and/or have total assets/fixed assets in the amount of the equivalent of up to EUR 43,000,000.

Large enterprises have more than 250 employees and have an annual turnover of the amount of the equivalent more than EUR 50,000,000, and/or have total assets/fixed assets in the higher amount of the equivalent of up to EUR 43,000,000.

Craft enterprise is a legal form of independent and permanent performance of permitted economic activities; production, turnover and the provision of services, in order to generate income or profit. It must have at least 1 employee. The holder of the craft enterprise, i.e., a physical person, is also the holder of responsibility with all his/her property.

Finally, total of 161 subjects have fulfilled all aforementioned requirements as the target research population. The distribution of the metal machining companies is shown on Figure 1.

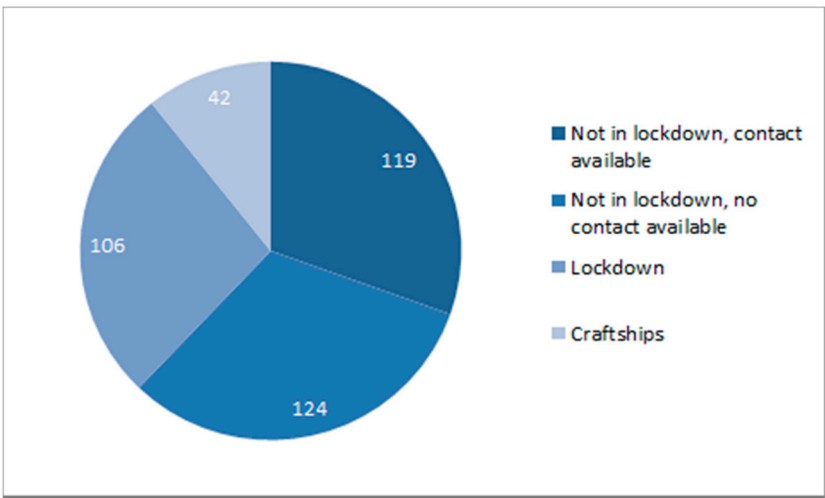

**Figure 1.** Metal machining companies in Croatia.

Data was collected by online questionnaire, structured in Google Forms online application. The goal was to perceive the general perspective of Industry 4.0 and its current application in the process planning sector. The questionnaire was divided into three groups of questions. The first group's goal was to collect general data about the company and the single participant/user. The participants were asked to enter the name of their company, their position within the company, level of education and familiarity with Industry 4.0 concept.

The second group of questions examines the use of the process planning methodology and its relation to the features of Industry 4.0. The requested information was a description of the current state in several elements of Industry 4.0. Those are the use of CAD and CAM, familiarity with CAPP, the use of databases, tools, machines, technology data archiving, time and cost estimation possibilities, the use of predictive analytics or special process planning algorithms, the existence of special intra-departmental communication channels, the use of lean tools and the tendency towards automation of processes. Participants had to offer a YES or NO answer, and also the data exchange frequency, intra-departmental communication level, hardware and software functionality, flexibility and capacity, organizational (de)centralization level and workers' motivation level were ranked on a scale from 1 (worst) to 5 (best).

In the third group, participants had to rank the importance of single elements of Industry 4.0 and its need for change within their own working environment. The elements were divided into three groups—hardware (1), software (2), organization and human resources (3). Elements of the hardware group were: hardware connection, sensors, servers, computer infrastructure, hardware system flexibility, hardware system modularity and hardware maintenance. Elements of the software group were: software connection, CAM, CAD, software maintenance, cyber-physical systems, database capacity, flexibility, decision support, software system modularity, social networks and self-optimization. Elements of the organization and human resources group were workers' motivation, workers' education, use of social networks, decentralization level and intra-departmental communication. The elements were ranked separately within each group.

The results were received from a total of 34 participants, which makes a total response rate of 21%. A total of 11 responses (32%) were received from small enterprises, 3 (9%) from medium, 1 (3%) from large, 13 (38%) from micro enterprises and 5 (15%) from craft manufacturing.

The participants were therefore grouped by their familiarity with Industry 4.0, level of education and company size. Familiarity with Industry 4.0 was defined by the answer to the question are they familiar with Industry 4.0 concept (yes/no). By company size, participants were grouped into small and medium enterprises group and micro and craft manufacturing group, as the first group tends to consist of larger companies and second tends to be smaller companies with about 10–20 employees.

By level of education, participants were grouped into those with middle and those with high education level. A middle educational level includes participants with high school education, and high education level includes participants with college (bachelor, master or PhD) education.

The results within groups were later tested, to acknowledge the difference between the groups by use of a non-parametric statistical test of the null hypothesis, the Mann-Whitney U test [47]. For the calculation of the priorities, the Friedman ranking test [48] was used. A total of 54% of the participants are familiar with Industry 4.0, 46% are not; 35% participants have a middle educational level and 65% have a high educational level, while 44% are from small and medium enterprises and 56% are from micro and craft manufacturing enterprises.

The Mann-Whitney U test is a nonparametric test of the null hypothesis that the probability that a randomly selected value from one population is less than a randomly selected value from a second population is equal to the probability of being greater. It will be used to test the differences between the groups where the null hypothesis states that there are no significant differences between the groups, while an alternative hypothesis states that there is a significant difference between the groups, that the i.e., company size, level of education and familiarity with Industry 4.0 all have an influence on the current state of process planning and several Industry 4.0 elements level of development in the company.

The Friedman test is a non-parametric statistical test. It is used to detect differences in treatments across multiple test attempts. The procedure involves ranking each row (or block) together, then considering the values of ranks by columns. The Friedman test is used for one-way repeated measures analysis of variance by ranks. In this paper, it will be only used to form the ranks of the Industry 4.0 elements by groups to perceive awareness of importance for their improvement and digitalization.

## 4. Results

The process planning in Industry 4.0, as previously described in reference [11], requires high level of work automation, use of predictive analytics and advanced CAPP systems. The development of process planning through industrial (r)evolutions, defined for this research and used as a reference point for the result evaluation, is shown in Figure 2.

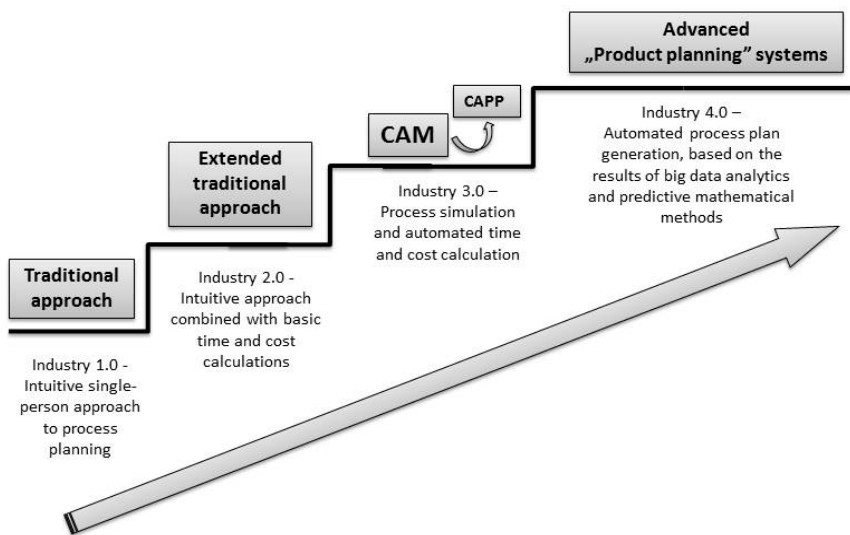

**Figure 2.** Industrial revolutions and process planning.

The traditional approach is based on the knowledge and experience of the single person, process planner, while process plan is being generated manually and intuitively, which can be related to Industry 1.0. The extended traditional approach uses the certain mathematical, yet relatively simple, methods in order to calculate preparatory, auxiliary and machining time, as well as the cost and,

for that reason, this is related to Industry 2.0. The use of CAM software can be related to the use of computers and work automation regarded to the 3rd Industrial revolution (Industry 3.0), which allows for the simulation of the process on a 3D model and automatic calculation of manufacturing time and cost. The next step in the industrial (r)evolution is the development of the CAPP systems, as a linkage between CAD and CAM, more advanced process plan generator, thus labeled as Industry 3.5. The final stage, designed by Industry 4.0 concept standards, is the "Product Planning" system, a completely digital, knowledge-oriented system that is directly connected with the manufacturing plan definition, uses predictive analytics and automatically creates the optimal process plan using the data from the archive, machine availability data and other various data collected from the entire supply and value chain [11].

### 4.1. Time and Cost-Estimation

The time and cost estimation is one of the most important steps in process planning, apart from the technology definition. This is the point where the subjectivity of a single user has to be minimized, while the use of mathematical methods is encouraged. In the digital environment of Industry 4.0, time and cost estimation is done automatically, with the aid of advanced mathematical methods (predictive analytics) and computer simulation. That is why the participants have been questioned regarding how they determine the manufacturing time and cost, and these results are shown in Figure 3.

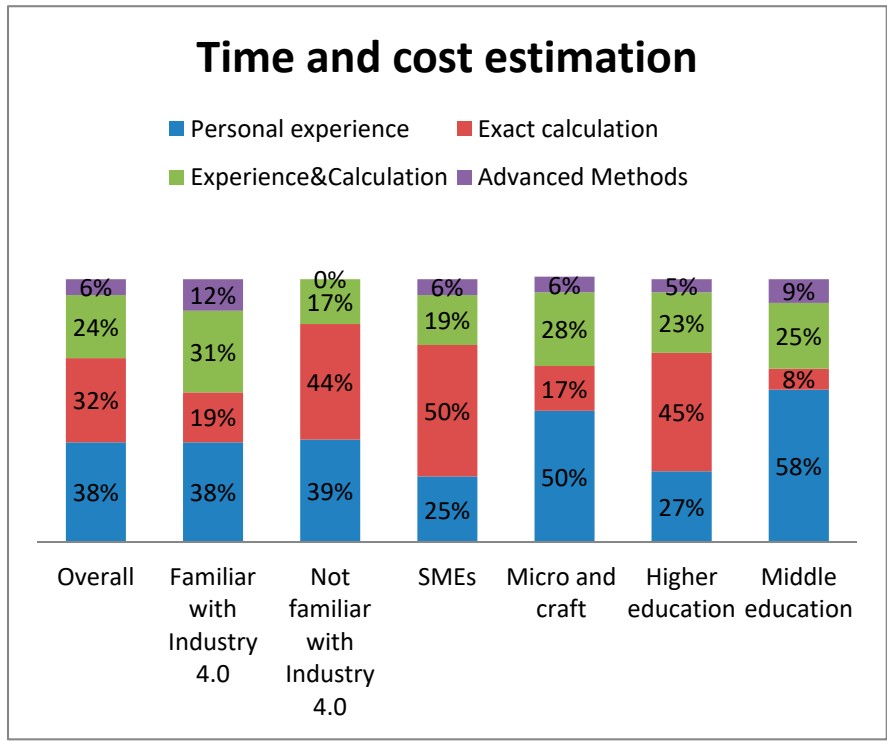

**Figure 3.** Time and cost estimation—overall.

In general, 38% of the participants are using the *traditional approach* and define the process plan only by their own intuition and experience, 32% use the exact calculation methods, while 24% use the combination of both and only 6% use advanced methods that are CAM simulation, a combination of intuition and CAM, while only one participant from each group (SMEs or micro/craft) has a customized advanced planning system.

The interesting finding, shown in Figure 3, is that a significant number of participants who claimed to be familiar with the term of Industry 4.0 still use *traditional approach* (38%), while 31% use the *combined approach*. A total of 12% of them use the advanced approach, which indicates that all of the

participants that use advanced simulation process planning are familiar with Industry 4.0. Those who still have not heard of Industry 4.0 mostly use the mathematical approach (44%), while 39% use the traditional approach (Figure 3).

When evaluating results and grouping the participants by the size of the company, it can be said that it was expected that 50% of the participants from micro and craft manufacturing use the traditional approach, while 50% of those in small, medium and big companies use the mathematical *approach*. Micro and craft manufacturers in Croatia, in most cases, have a single person, usually an owner, who does multiple tasks, as well as process planning, while mostly relying on their own experience. As mentioned above, a single participant that uses advanced methods has been identified in each group, i.e., the SMEs group and the micro/craft group. The results have, therefore, shown that the size of the company is in correlation with the process planning approach—micro and craft manufacturing tend to use the traditional approach, while SMEs are more based on the the extended traditional approach.

As for the participants' educational level, the results have shown that 45% of those with high level of education use the mathematical approach (extended traditional), while 58% of those with a middle educational level use the traditional approach, which is the biggest and also expected difference between those two categories examined. The 88% of the participants from SMEs have a high educational level, and only 12% middle. In the micro and craft manufacturing group, only 44% have a high educational level while 64% have a middle educational level, which explains the correlation between increased use of the traditional approach in micro and craft manufacturing and educational level. Finally, it can be concluded that the level of education is not an obstacle in terms of the level of advancement of methods used.

## 4.2. Use of CAD-CAM and CAPP

The first step of process planning digitalization is the use of CAD, with the possibility of 2D drawing manipulation, while the second represents the usage of CAM enabling 3D modelling of the product and advanced process simulation. CAPP is a linkage between CAD and CAM, which enables the automatic generation of the process plan and is a part of process planning 3.5, one step from complete digitalization of product planning systems. CAPP is rarely used in Croatian manufacturing, which is also proven by the obtained results in the field of metal machining, wherein only 44% of participants have heard of CAPP (Figure 6). The results have also indicated that most of the participants (88%) use CAD software for process planning (Figure 4).

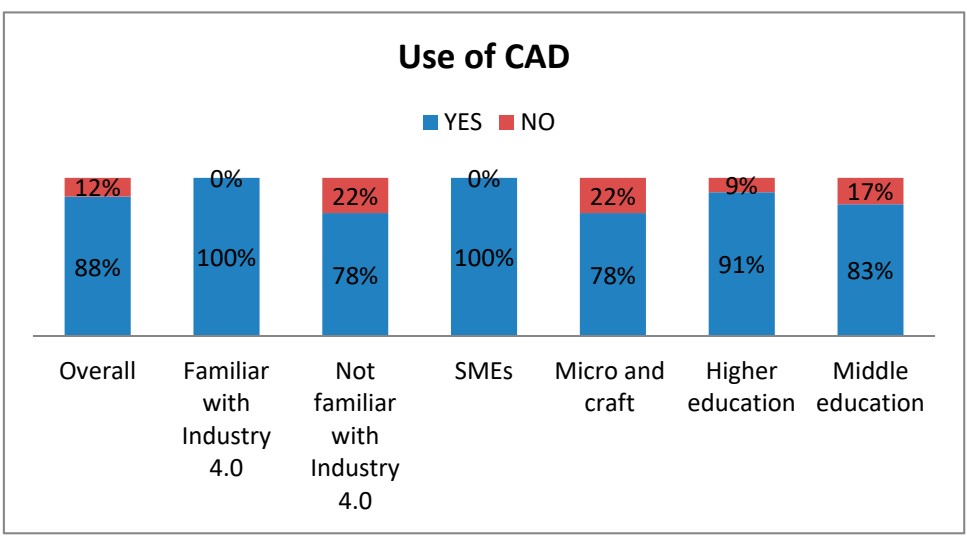

**Figure 4.** Use of computer-aided design (CAD) software.

As shown in Figure 4, use of CAD software is omnipresent and all participants who are familiar with Industry 4.0 use CAD, while 78% of those who are not familiar with Industry 4.0 use it nevertheless. There is high level of CAM software usage (Figure 5) present as well, among 76% of the entire population, while 94% who are familiar with industry 4.0 use CAM and 61% who are not familiar with Industry 4.0 use CAM. CAPP is mostly unknown to the participants, with only 44% having heard of it, out of which, 31% are familiar with Industry 4.0 and only 22% are not (Figure 6). A total of 69% of those familiar with Industry 4.0 are familiar with CAPP, and only 22% of those who are not familiar with Industry 4.0 are familiar with CAPP systems.

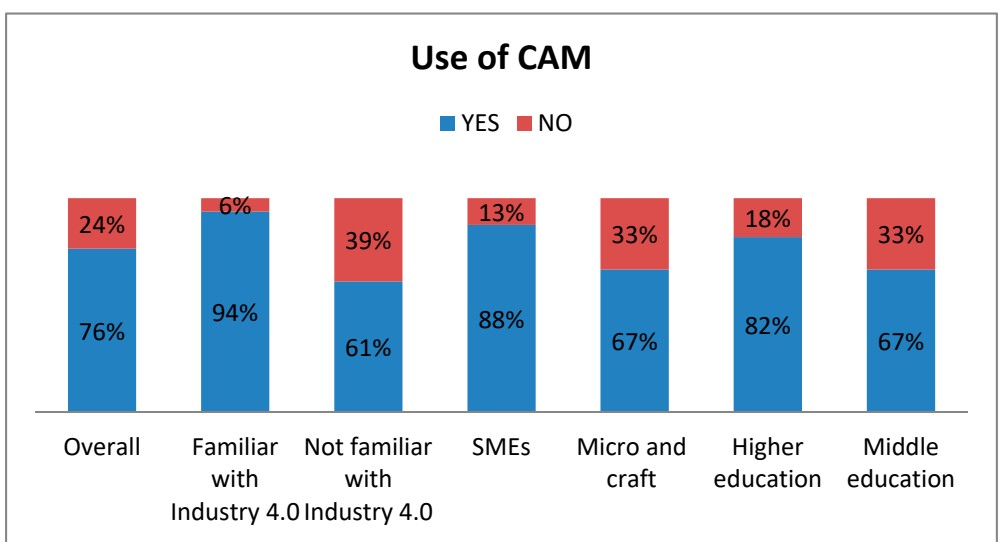

**Figure 5.** Use of computer-aided manufacturing (CAM) software.

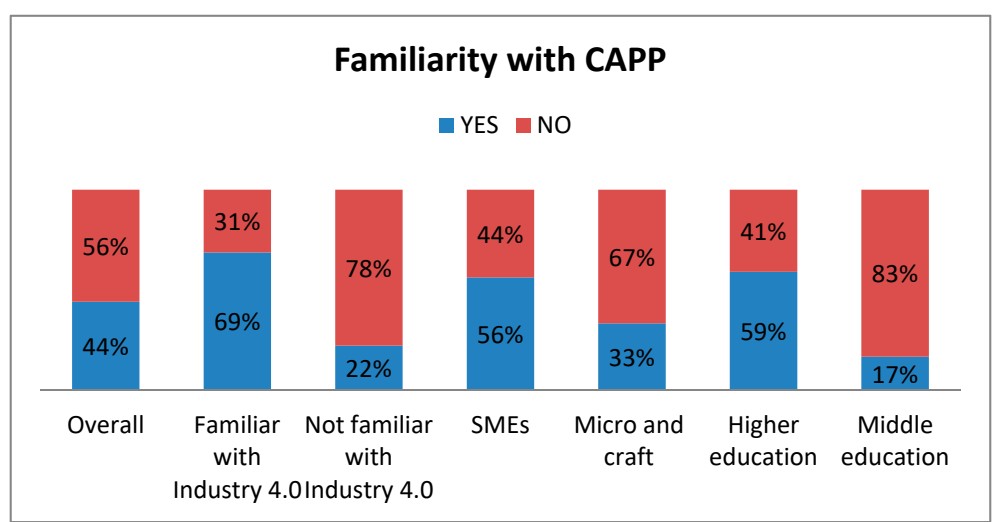

**Figure 6.** Familiarity with computer-aided process planning (CAPP).

Regarding the educational level of participants, the most significant difference is that those with a middle educational level are not as familiar with CAPP systems as are those with a higher educational level (59% vs. 17%). Additionally, the use of CAM is most significant among those with a higher educational level (82% vs. 67%) (Figure 5).

When considering the size of the company, 100% of SMEs and 78% of micro and craft enterprises use CAD (Figure 4). Additionally, the use of CAM is in ratio 88% vs. 67% (SMEs/micro-craft) (Figure 5). A total of 56% of SMEs are familiar with CAPP, while only 33% of micro and craft enterprises are familiar with the same system (Figure 6).

### 4.3. Existance of Big Data in Databases

One of the most important characteristics of the Industry 4.0 is big data analytics, which demands the existence of adequate databases, internet infrastructure and familiarity with predictive analytics.

Considering the availability of databases with information about processes, machines, tools and other useful data for process planning, 68% of participants have responded that there is already an existing database available in their company (Figure 7). A total of 75% of those familiar with Industry 4.0 have an existing database, as well as the 61% of those who are not. There is no significant difference in the educational level of participants regarding the database availability, while there are more (75% in particular) databases available in the case of SMEs, than in the micro and craft manufacturing (61%).

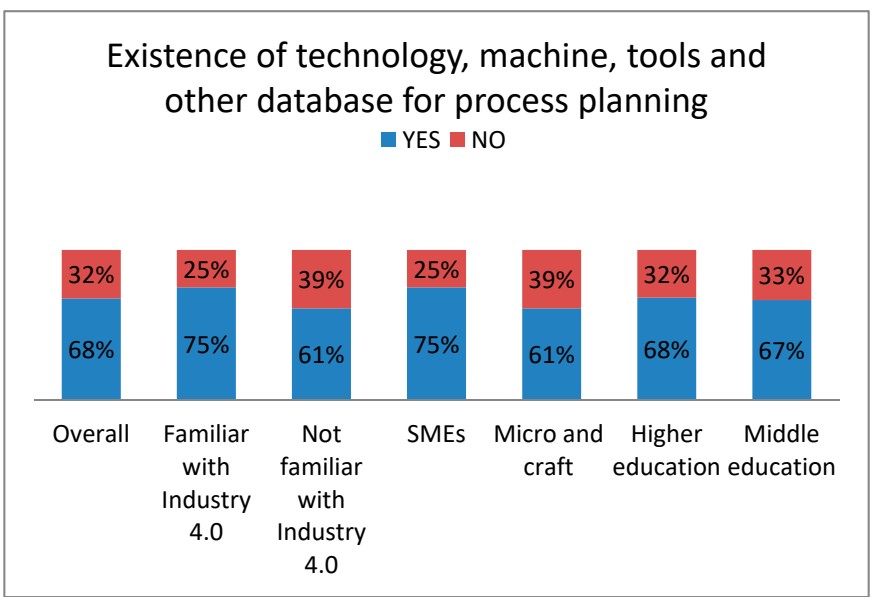

**Figure 7.** Database presence and development.

### 4.4. Machine Learning and Predictive Analytic Methods

The use of advanced mathematical methods enables the process planning digitization, while, at the same time, requires the use of the data from the archive. These data are processed, depending on the level of digitization, by means of specific algorithms or predictive analytics methods that enable the machine learning which, finally, results in the automatic generation of an optimal process plan.

As shown in Figure 8, most of the participants (82%) have acknowledged that they are using data from an archive to create the process plan. Only 3% use a specific algorithm and 15% use the predictive analytical methods. Hence the small response rate on algorithms and algorithm usage, the significant difference in groups regarding size and education has not been noticed, the participants are evenly distributed within each group.

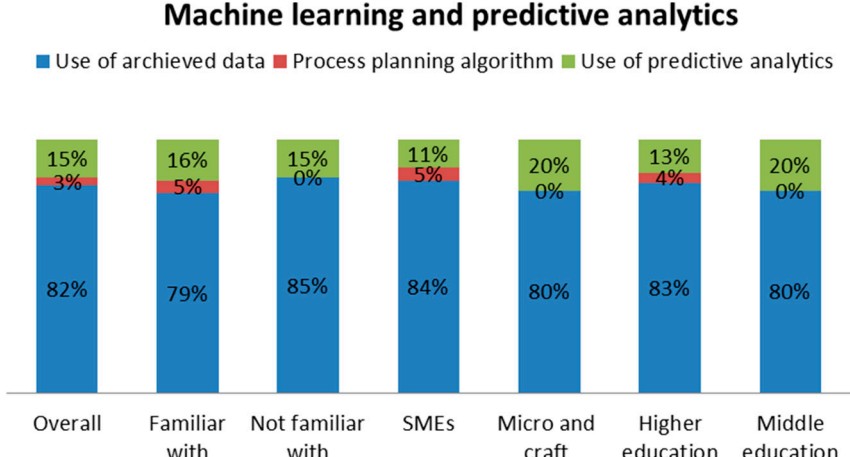

**Figure 8.** Machine learning and predictive analytics.

### 4.5. Hardware and Software Capacity

In order to facilitate the digital company concept, the hardware and software infrastructure have to be functional on a very high level. That is why the participants have been questioned about their satisfaction with the existing internet infrastructure, software flexibility and its capacity and about functionality and capacity of the hardware used in the company. The participants have evaluated the activity on a scale from 1 (extremely bad) to 5 (extremely good), and the results are shown in Figure 9. The average grade in every group has been calculated and the significance of difference between each groups has been examined with the Mann-Whitney U-test, with a 0.95 confidence interval, where the hypothesis has been defined that there is no difference between the answers in the groups. If not, the alternative hypothesis claims that there is a significant difference between the answers in the groups noticed [16].

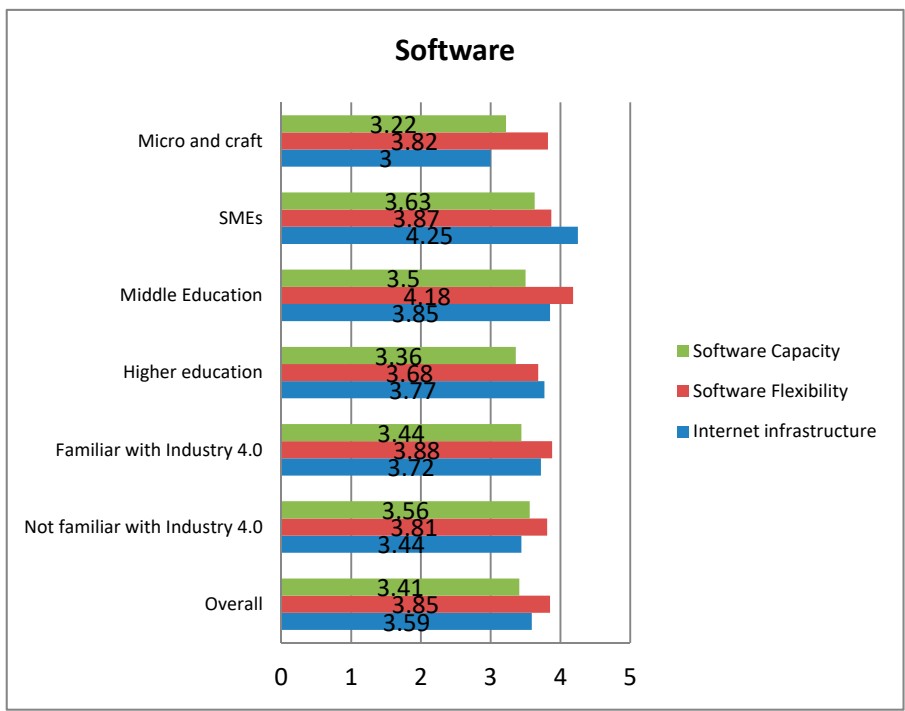

**Figure 9.** Software evaluation.

The average rating of internet infrastructure is 3.59, software flexibility 3.85 and software capacity 3.41 (Figure 9).

The most noticeable difference in the results can be seen in SMEs, which have highest rate of internet infrastructure functionality (4.25). The significance of the difference from another group (micro-craft) in this field has been confirmed by the alternative hypothesis with $p = 0.0017$, which implies a large and very distinctive difference within two groups.

Those with a middle educational level have ranked flexibility of their software the highest (4.18).

The familiarity with the Industry 4.0 concept has not indicated any significant variations in responses regarding software evaluation.

The functionality of hardware and hardware capacity is also evaluated on a scale of 1 to 5 (Figure 10). The hardware functionality gained average grade of 3.82, while capacity gained average grade of 3.59 among the participants. Again, the familiarity with Industry 4.0 has not shown the difference in the hardware infrastructure responses, nor the educational level, but the significant difference can be noticed in the company size—SMEs are more satisfied with hardware functionality (4.19) and capacity (4.13), and the micro and craft manufacturing companies are shown to be less satisfied with their hardware capacity (3.11), which has been proven with $p = 0.002$ and $p = 0.0063$. With $p = 0.05$ as a statistical limit value, it can be concluded that the differences in these two cases are large.

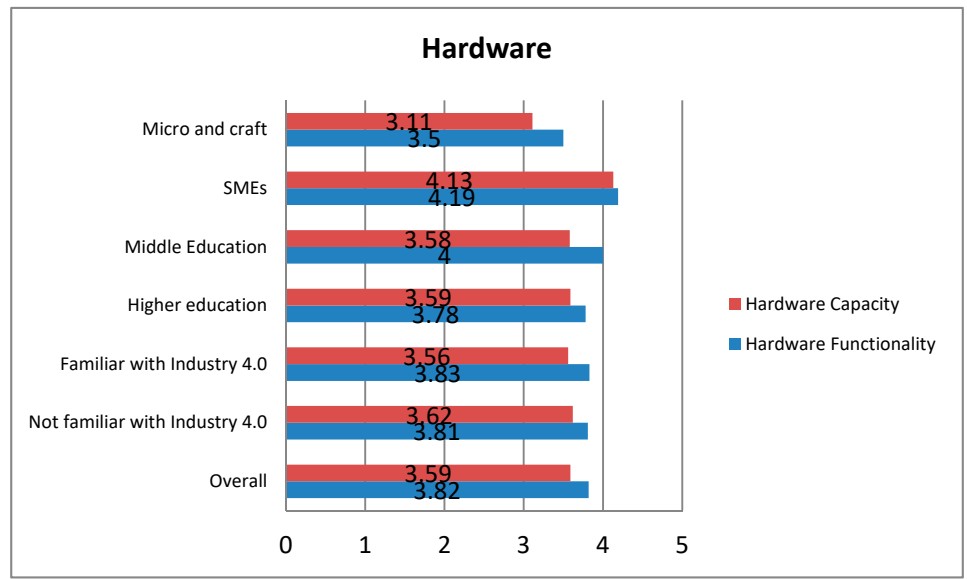

**Figure 10.** Hardware evaluation.

### 4.6. Communication within the Value Chain

The Industry 4.0 concept demands both horizontal and vertical integration. Horizontal integration requires the connection to every part of the value chain from which data is being collected, in order to acknowledge its impact on the process planning with future use in the optimization of the local processes. The participants were asked to evaluate their communication level with several parts of the value chain: product design, suppliers, customers, planning and manufacturing and other departments within their company (Figure 11).

The results have shown that the process planning department has most frequent and most effective communication with the product design and planning/manufacturing department. This result was expected, since those are directly connected within the value chain. In particular, product design precedes the process planning and manufacturing follows.

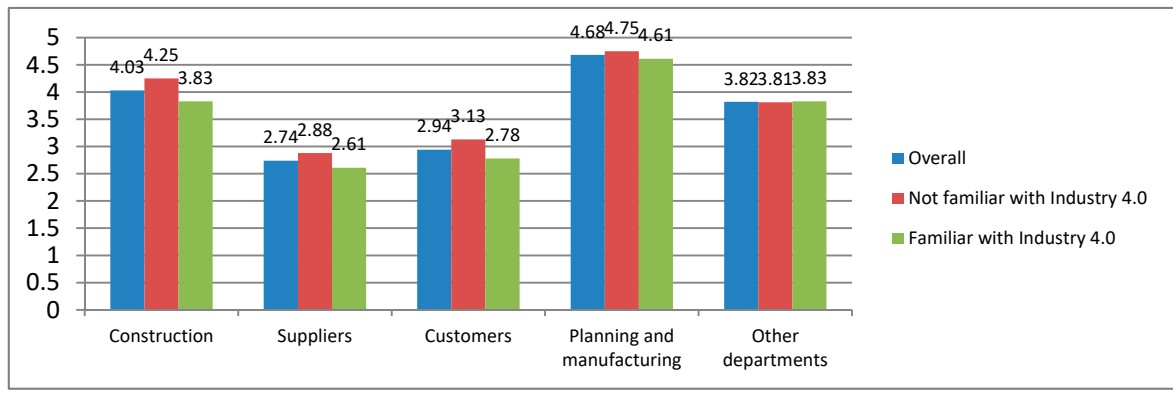

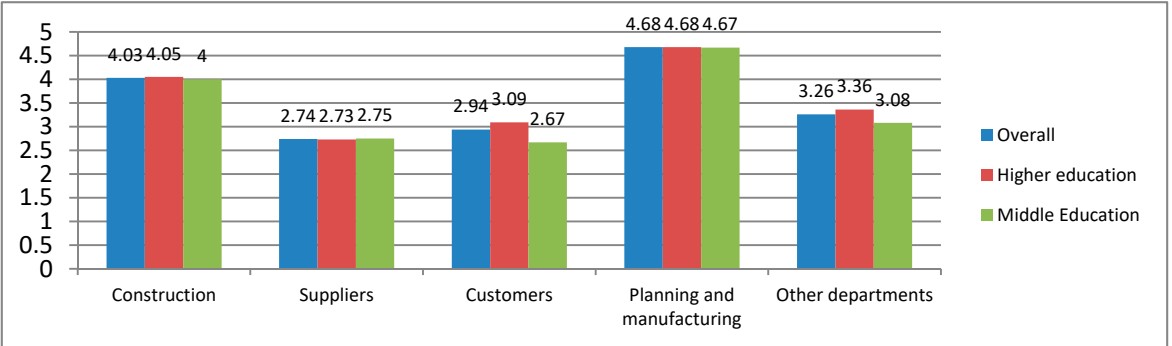

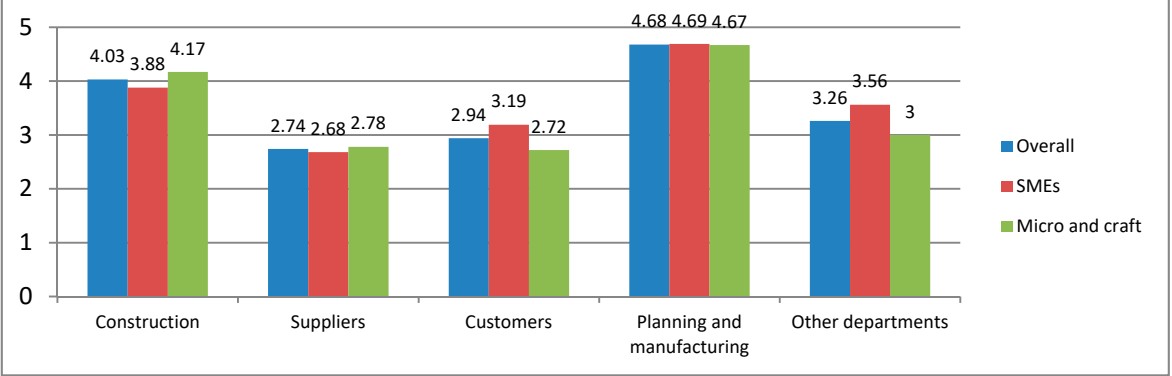

**Figure 11.** Communication in the value chain.

Process planners tend to have the lowest rate of connection with suppliers (rated 2.74) and the customers (2.94). The ones unfamiliar with the Industry 4.0 concept have lowest rating in communication with the suppliers (2.61) as well as the customers (2.78).

The significant difference in the ratings has been noticed regarding connection with other departments within the company based on the company size. SMEs have rated their connection as 3.56 while the micro and craft manufacturing have rated theirs as only 3 ($p = 0.0455$). This value is close to the statistical significance limit of $p = 0.05$, which leads to conclusion that the difference is not as radical, but yet it exists as such.

### 4.7. Impact of the Human Factor

The traditional process planning approach is characterized by a high level of human subjectivity. That is why the participants have been questioned on the judgment on their current level of subjectivity during the process planning, but also about the possibility of their impact and flexibility in the working activities. Additionally, they were asked to evaluate the possibility of their impact, as process planners, on the change and implementation of certain modifications in the working environment. Flexibility

therefore considers the possibility to approach every case differently with ease, while impact of change considers their impact on the working environment improvement (Figure 12).

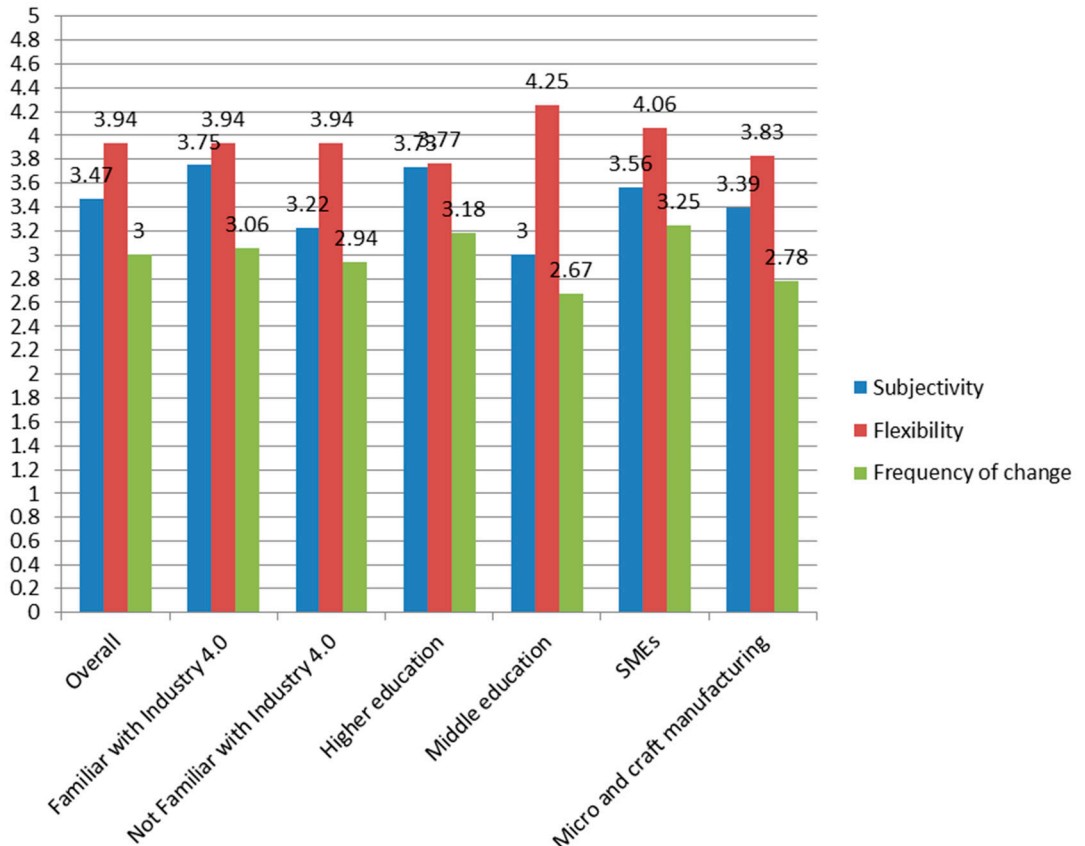

**Figure 12.** Impact of the human factor.

Since the subjectivity in the digital era should be minimized, it can be concluded that the rate of 3.47 in general is very high, especially within those groups that are familiar with the Industry 4.0 concept, which are shown to have the highest rate of subjectivity level (3.76). Additionally, it is interesting that those with higher education estimate that their subjectivity level at 3.73, while those with a middle educational level have the lowest perception of their subjectivity rate (being 3 in that particular case, with $p = 0.0369$).

On the other hand, participants consider that they have a possibility for high flexibility, with rating of 3.94 on average, while the rate is the highest among those with a medium educational level (4.25) and in those working in SMEs (4.06).

According to the results, participants apparently are not open to continuous change, which got the lowest rating within those categories (3). Those with a medium educational level have achieved the lowest rating (2.67), while those working in SMEs rated the highest (3.25).

*4.8. Improvement Priorities*

In the third part of the questionnaire, the participants were asked to rank the certain enterprise elements (dimensions) by the improvement priority. Table 1 shows the overview of the dimensions, while the comparison of the ranks evaluated by Friedman test is shown in Table 2.

**Table 1.** Industry 4.0 elements and its improvement priorities.

| Hardware | | Software | | Organization and HR | |
|---|---|---|---|---|---|
| Element | Code | Element | Code | Element | Code |
| Hardware Connection | H-1 | Software connection | S-1 | Worker motivation | O-1 |
| Sensors | H-2 | CAM | S-2 | Worker education | O-2 |
| Computer infrastructure | H-3 | CAD | S-3 | Use of social networks | O-3 |
| Servers | H-4 | Software maintenance | S-4 | Decentralization | O-4 |
| Hardware system flexibility | H-5 | CPS | S-5 | Intra-departmental communication | O-5 |
| Hardware system modularity | H-6 | Database capacity | S-6 | | |
| Hardware maintenance | H-7 | Software flexibility | S-7 | | |
| | | Decision support | S-8 | | |
| | | System modularity | S-9 | | |
| | | Social networks | S-10 | | |
| | | Self-optimization | S-11 | | |

**Table 2.** Friedman ranking test of Industry 4.0 elements by groups.

| HARDWARE | | | | | | | | | | | | | |
|---|---|---|---|---|---|---|---|---|---|---|---|---|---|
| Overall | | Familiar with I40 | | Not Familiar with I40 | | SMEs | | Micro and Craft | | Higher Education | | Middle Education | |
| H-5 | 3.3333 | H-5 | 2.7500 | H-3 | 3.1875 | H-2 | 3.1786 | H-5 | 3.5667 | H-5 | 3.3333 | H-5 | 3.3333 |
| H-3 | 3.6667 | H-1 | 3.6071 | H-4 | 3.7813 | H-5 | 3.2143 | H-7 | 3.5667 | H-3 | 3.5000 | H-6 | 3.6111 |
| H-1 | 3.9333 | H-6 | 4.1071 | H-5 | 3.8438 | H-1 | 3.6786 | H-4 | 3.7000 | H-4 | 3.6667 | H-1 | 3.6667 |
| H-4 | 3.9500 | H-3 | 4.1429 | H-6 | 3.8750 | H-6 | 4.0714 | H-1 | 4.0000 | H-1 | 4.0476 | H-7 | 3.7222 |
| H-6 | 3.9833 | H-4 | 4.2143 | H-7 | 3,9375 | H-4 | 4.2143 | H-6 | 4.0667 | H-6 | 4.1429 | H-3 | 4.0556 |
| H-7 | 4.2167 | H-7 | 4.5357 | H-1 | 4.2188 | H-7 | 4.7500 | H-3 | 4.1000 | H-7 | 4.4286 | H-4 | 4.6111 |
| H-2 | 4.9167 | H-2 | 4.6429 | H-2 | 5.1563 | H-3 | 4.8929 | H-2 | 5.0000 | H-2 | 4.8810 | H-2 | 5.0000 |

| SOFTWARE | | | | | | | | | | | | | |
|---|---|---|---|---|---|---|---|---|---|---|---|---|---|
| Overall | | Familiar with I40 | | Not Familiar with I40 | | SMEs | | Micro and Craft | | Higher Education | | Middle Education | |
| S-1 | 3.9444 | S-1 | 4.0385 | S-1 | 3.8571 | S-1 | 3.3750 | S-1 | 4.2857 | S-2 | 3.9737 | S-1 | 3.4375 |
| S-2 | 4.6111 | S-2 | 4.8077 | S-2 | 4.4286 | S-2 | 4.7917 | S-2 | 4.6071 | S-1 | 4.1579 | S-7 | 4.3125 |
| S-7 | 4.9074 | S-3 | 4.8077 | S-7 | 4.6786 | S-7 | 5.0417 | S-7 | 4.9643 | S-3 | 4.6316 | S-8 | 5.0625 |
| S-3 | 5.2222 | S-7 | 5.1538 | S-3 | 5.6071 | S-3 | 5.6250 | S-3 | 5.0714 | S-7 | 5.1579 | S-6 | 6.0000 |
| S-4 | 6.0926 | S-4 | 5.5000 | S-6 | 6.2143 | S-11 | 5.6250 | S-4 | 5.5714 | S-4 | 5.9737 | S-2 | 6.1250 |
| S-6 | 6.3704 | S-8 | 6.3462 | S-10 | 6.2143 | S-8 | 6.1667 | S-6 | 6.3929 | S-9 | 6.2368 | S-4 | 6.3750 |
| S-8 | 6.4074 | S-6 | 6.5385 | S-8 | 6.4643 | S-9 | 6.1667 | S-5 | 6.5000 | S-11 | 6.4211 | S-10 | 6.4375 |
| S-11 | 6.5556 | S-9 | 6.7692 | S-4 | 6.6429 | S-6 | 6.3750 | S-8 | 6.6429 | S-6 | 6.5263 | S-3 | 6.6250 |
| S-9 | 6.7778 | S-11 | 6.9231 | S-9 | 6.7857 | S-4 | 6.4167 | S-11 | 7.1429 | S-8 | 6.9737 | S-5 | 6.6875 |
| S-5 | 7.1481 | S-5 | 7.2692 | S-5 | 7.0357 | S-5 | 7.7083 | S-10 | 7.2143 | S-5 | 7.3421 | S-11 | 6.8750 |
| S-10 | 7.9630 | S-10 | 7.8462 | S-11 | 8.0714 | S-10 | 8.7083 | S-9 | 7.6071 | S-10 | 8.6053 | S-9 | 8.0625 |

| ORGANIZATION AND HUMAN RESOURCES | | | | | | | | | | | | | |
|---|---|---|---|---|---|---|---|---|---|---|---|---|---|
| Overall | | Familiar with I40 | | Not Familiar with I40 | | SMEs | | Micro and Craft | | Higher Education | | Middle Education | |
| O-2 | 2.2424 | O-2 | 2.0625 | O-1 | 2.3235 | O-2 | 2.3667 | O-1 | 2.1765 | O-1 | 2.3409 | O-1 | 1.8636 |
| O-1 | 2.2576 | O-1 | 2.1875 | O-3 | 2.4118 | O-1 | 2.4000 | O-2 | 2.1765 | O-2 | 2.4545 | O-2 | 2.0455 |
| O-5 | 2.7424 | O-5 | 2.,8750 | O-5 | 2.6176 | O-5 | 2.6000 | O-5 | 2.8529 | O-5 | 2.8636 | O-5 | 2.5000 |
| O-4 | 3.7576 | O-3 | 3.9375 | O-4 | 3.5882 | O-4 | 3.6667 | O-4 | 3.7941 | O-4 | 3.5682 | O-4 | 4.1364 |
| O-3 | 4.0000 | O-4 | 3.9375 | O-2 | 4.0588 | O-3 | 3.9667 | O-3 | 4.0000 | O-3 | 3.7727 | O-3 | 4.4545 |

Within the hardware group, the highest priority for change was the hardware flexibility dimension, followed by computer infrastructure and hardware connection (Table 2). While considering priorities in certain groups, SMEs have given the highest priority to sensors, and most of the other groups consider this dimension as the lowest priority for change. Micro and craft manufacturing have given second high importance to hardware maintenance, while others do not consider this dimension as the one with the highest importance. Those unfamiliar with the Industry 4.0 concept have given one of the lowest priorities to hardware connection, which can be related to their lack of knowledge of the concept that demands high level of connectivity on every level.

In the software group, in most cases, the highest priority was given to software connectivity, followed by CAM systems and software flexibility. The improvement of CAM systems leads to digitalization of the process planning, and is one step closer to CAPP achievement. The lowest importance was given to social networks that enable intra-departmental communication and cyber physical systems, which may be more complex to implement and are not directly connected to process planning as such. Radical differences in ranking between the groups have not been noticed.

When considering the organization and human resources group, every group gave the highest importance to worker motivation and education, while the lowest importance was given to decentralization and use of social networks for local communication.

## 5. Discussion

The results have shown that process planning in Croatia is still highly affected by the traditional approach, which is based on a single-person intuition, knowledge and subjectivity. A total of 38% of the participants make their decisions by experience only, while 32% of them do the exact mathematical calculations, which can be acknowledged as a slightly advanced mathematical approach. A total of 26% use the combined approach and only 6% the advanced approach, which can be a first step of the digital process planning towards Industry 4.0 implementation.

The familiarity with Industry 4.0 concept of the participants does not make a significant difference on the process planning method used, as there also has not been any significant difference noticed in the user satisfaction about the working environment. The only correlation that has been noticed is the fact that those who are not familiar with Industry 4.0 concept in general are not familiar with CAPP systems.

When the participants have been grouped by company size (wherein companies have been categorized into two groups, i.e., SMEs and micro-craft manufacturing), a significant difference has been acknowledged in the internet infrastructure rating. SMEs have labeled their internet infrastructure with the grade of 4.25 and micro-craft manufacturing with only 3. Additionally there is a difference in hardware capacity: SMEs have labeled their hardware capacity with 4.13, while micro/craft with 3.11. The internal communication functionality differs by the size of the company: SMEs have labeled it with 3.56, while micro-craft with 3. A total of 80% of SMEs have machining, auxiliary and preparatory time archived, while the same can be found in only 44% micro-craft manufacturing.

SMEs tend to use the mathematical approach in process planning, which can be attributed to the longer company tradition and work activities standardization.

The level of education of the participants has an influence on the current digitization level. There is significant difference in subjectivity. Those with higher education have estimated influence of subjectivity in their work with 3.78 rating, while those with a middle educational level have estimated their susceptibility to being subjective with the rating of 3. It can be said that this is quite unexpected, and that those with higher education are expected to have lower rate of subjectivity in their activities. A total of 60% of those with higher education are familiar with CAPP, while only 16% of those with a middle educational level are familiar with this advanced system, which has actually been expected.

As there have not been any radical results regarding the infrastructure, hardware and software, as well as the organizational environment, participants still have not considered the improvement of

their work activities. Nobody seems to use specialized algorithms for process planning automation, with only a few exceptions of predictive analytics being used (6%).

The complete overview of the Industry 4.0 elements in order by their priority rates examined in the previous chapter with their average grade given (1–5) or percentage of use, target goal and actions to be taken for the improvement, are shown in Table 3.

**Table 3.** Industry 4.0 elements and its perspective.

| Industry 4.0 Element | Average Grade or Percentage of Usage | Target | Actions to Be Taken |
|---|---|---|---|
| *Hardware* **(priority level: 2.2333)** | | | |
| Computer infrastructure | 3.59 | One of the most important characteristics of Industry 4.0, must always be functional, at high speed and by the latest market trends. | Increase the Internet infrastructure at the highest level available in the company area. Discuss with the local Internet providers and regional government the install and development of better Internet infrastructure as the key Industry 4.0 element. |
| Hardware functionality | 3.82 | Functionality has to be on the highest possible level, enabled by the pro-active maintenance methods so that there without need for extra time calculation regarding the functionality. | Increase the hardware functionality, need for modern hardware components use. |
| Hardware capacity | 3.59 | Hardware has to have a functional capacity, which is adaptive and flexible, equally functional in the need of unexpected decrease and increase of the demands. | Increase the hardware capacity. |
| *Software* **(priority level: 2.1667)** | | | |
| Cloud Computing | 12% | Cloud-based processes, every software action is being developed and performed in the cloud. Data, information and knowledge is also stored and processed in the cloud. | After the proper Internet infrastructure development, the cloud computing will be possible and effective. |
| Software connectivity | 4.18 | Every part of supply chain is software-connected and can exchange the data, information and knowledge. | Mild improvement of software connectivity can be handled with providing the better Internet infrastructure. |
| Flexibility | 3.85 | Enables the fast changes regarding different needs and new market trends. Shortens the transformational period time and enables the stable and innovative functionality. | Due very low software flexibility, its increase should be taken as very important and priority factor. |
| CAM | 76% | Omnipresent use of CAM as part of advanced software systems. | Upgrade the use to 100%. |
| CAD | 88% | Omnipresent use of CAD as part of advanced software systems. | Upgrade the use to 100%. |
| Software capacity | 3.5 | Capacity must be flexible and functional, unlimited. | Increase the software capacity at the level of accurate functionality and upgrade possibilities. |
| Database and archive existence | 68% | Databases with archived data from the past are essential for the building the predictive behaviour. They must be functional and user-manipulative. | Increase the development of databases and process plan archives, in order to use it for the future prediction and automatic process plan generation. |

**Table 3.** *Cont.*

| Industry 4.0 Element | Average Grade or Percentage of Usage | Target | Actions to Be Taken |
|---|---|---|---|
| Predictive analytics | 18% | Prediction of the need for tools, machine availability, raw material and optimum use of technologies. | Increase the use of predictive analytics in process planning; in the further digitalization stages make it a standard tool. |
| CAPP | 44% | CAPP as base of "Product planning" software. | Educate the companies about the CAPP, its role and possibilities, implementation of the CAPP in the work environment. |
| Self-Optimization | 0 | Based on the data collected from the various parts of the value chain, every new generated process plan is being more optimal than the previous one. | First step is to make an audit how and where to use self-optimization systems. |
| Database capacity | 68% Excel, 3% Access, 6% CAM, 6% Advanced ──────── 68% Excel | Flexible database capacity enables the continuous change of the data needed and enables its functionality. | Avoid Excel use, develop the customized online databases which enable better data manipulation, are connected with other parts of value chain and enable the team work. |
| Big data manipulation | 6% | Effective, easy, accurate and fast. | Educate the companies about the big data and its use, the possibilities and advantages of its collection and manipulation. Definition of big data role within the current and future system, possible the correlation with KPIs. Development of the customized system for big data analytics and customization. |
| Use of Social Networks | 42% | Fast, accurate and secured intra-departmental communication. | Broaden the use of internal social networks for easy and secure human-human communication within the organization. |
| *Organization and human resources* **(priority level: 1.6000)** | | | |
| Worker education | 47% | Fully educated in new digital technologies, control mechanisms and system optimization, lifelong learning process. | Increase the worker education level in the basic knowledge of Industry 4.0, use of digital technologies and their new role within the organizational system. |
| Worker motivation | 4.59 | Motivation must be at highest level possible because of the radical digitalization changes as well as the unpredictable working environment. | The worker motivation is high enough to start the transitional process; motivated workers help those who still do not see the benefits of the digitalization. |
| Intra-departmental communication | 3.64 | Fast, accurate and secured intra-departmental communication. | Increase the communication level, collect data in real time from other parts of organization. |
| (De)Centralization | 3.72 | Complete and effective horizontal and vertical integration. | Level of centralization in the company is yet very high and has to be decreased on the lowest level possible. |
| Human subjectivity | 3.47 | As minimized as possible. | Minimize the level of human subjectivity, as now is very high. First step is implementation of exact mathematical methods in process planning and second is the use of CAPP and automatic generation of the process plan. |

**Table 3.** *Cont.*

| Industry 4.0 Element | Average Grade or Percentage of Usage | Target | Actions to Be Taken |
|---|---|---|---|
| Human flexibility | 3.94 | At highest level due innovative approach and stable motivation level. | Mild increase is needed, as the human flexibility also has to be on very high level. |
| Environment of change | 3 | Unpredictable market and working environment, human is adaptive to change and embrace it as the usual. | Human within the organizations are not yet as adaptive to change as they should be. The changing and unstable environment is to be taken as normal. Education of the workers is needed. |
| Lean tools | 12% | Complete use of lean tools and kaizen philosophy of continuous improvement in work. | Priority of increasing of lean tool uses, make a plan of lean transformation and make sure that transitional period is being performed by the lean and kaizen standards. |

Software flexibility and connectivity have been rated with the highest priority for future change within the software group. While the software connectivity is evaluated with average grade of 4.18, only 12% of users are familiar with cloud computing, a feature that is an essential part of software connectivity. Up next in the priority for change are CAD and CAM, wherein a high level of usage can already be noticed. The CPS have been rated at the bottom of the priority for change: while only 6% of users are familiar with big data manipulation, only 18% use predictive analytics, with only 6% of users having unique databases developed for their needs. Satisfaction with software flexibility level is also relatively high, rated with 3.85, and, at the same time, this segment is recognized as the top priority for change (Table 3).

Hardware flexibility and servers are rated with highest change priority in hardware group, while hardware functionality has been given the average grade of 3.82, which is also a relatively high average grade. Internet functionality and hardware capacity have been graded with 3.59, as servers and their capabilities are rated as top priority for change.

Organization and human resources have been recognized with the highest priority for change and, at the same time, the human motivational level is graded as very high (4.59). There are certain difficulties to be resolved within the communication channels in the value chain, but, apart from that, based on these results, the conclusion can be made that users have rated the features which function the best within their working environment as the top priority for change.

Research about the process planning in Industry 4.0 in the metal machining industry is a novelty in the field, so it can be only roughly compared to the previously conducted empirical research and evidences from the practice from other countries, which is why only basic comparison of most recent similar studies will be given. Such research, conducted among Danish companies, has shown that the lack of knowledge does not reduce Industry 4.0 readiness of their SMEs [49], which is opposite to Croatian SMEs, which consider the increase of workforce education as one of the most important elements in the transitional period. The difference is also found with respect to work motivation, where, in Croatia, this is evaluated as being at a very high level, while, in Denmark, it is considered to be one of the most challenging readiness elements. On the other hand, the correlation of lack of concept understanding and readiness has been found in Danish companies, which is similar to the Croatian metal machining companies, wherein 46% of the participants still have not even heard of Industry 4.0 [16]. In comparison to the research conducted in Serbia, wherein digitally-mature companies are assessed, human resources are perceived as an obstacle to Industry 4.0 implementation, because of the lack of necessary competences and skills, which is also very similar to Croatian companies. That particular study also mentions that the main impact can be seen in increasing manufacturing process efficiency, which can be related to the highest importance being given to hardware and

software connectivity and flexibility, which increases process efficiency [49]. In Italian companies, the highest importance was given to cloud computing technologies, ICT integration (machinery, electronic equipment and database), but also the use of new technologies (i.e., additive manufacturing), has shown to be one of the priorities for digital transformation strategy [50], while another research (also from Italy) has shown that many of the participants from the industry have no knowledge of cloud-based technologies and the use of augmented and virtual reality in the working environment, which is similar to Croatian companies [51]. Research conducted in Polish companies also implied high importance of investment in new hardware and software technologies, but also in human resources [52], while research from South Africa has shown a similar result of importance in innovativeness level, but also the difference in big data system development, where their result has shown very high awareness of the benefits from this kind of data processing, unlike Croatian companies, who are still mostly not aware of importance of this key Industry 4.0 element [53].

Regarding data collection and processing, the study conducted in Czech Republic manufacturing companies enhanced the importance of its implementation, giving it the highest importance factor, but also in the general view, a very similar conclusion has been made that the most of the companies still have not implemented the concept at the fullest, and that there is a high level of diversity between the implementation of certain elements, where some can be fully implemented by the Industry 4.0 standards, while others are not being used or even considered for implementation [54]. This research has found similar results within Croatian companies who tend to use, e.g., CAM software, widely, but most of them still are not even aware of the existence of CAPP systems.

## 6. Conclusions

The traditional approach in process planning, based on a single person's intuition, experience and knowledge, is still widely used in Croatian companies. From the research in metal machining companies, it can be concluded that the majority prefer to accept digitalization as a natural state of development, rather than a need for urgent transformation in order to remain in the leading position of the market.

One of the main challenges during the transformational period is the education of the workers because their familiarity with Industry 4.0 or even the CAPP concept still is not at a satisfying level, which is the first step to start to plan the digitization strategy. The motivational level is high and the human and organizational component of the company is acknowledged as the most important to users. Results regarding hardware, software and internet infrastructure have shown the middle functionality level on average, while most of the work is still spreadsheet-based. Only few participants have shown certain actions taken towards the complete digitalization of the process planning. The traditional approach and manual mathematical approach are yet the most frequently used within the Croatian industry, which has a motivation to change, but those changes need to be radical, regarding the current under-developed state of this particular sector in terms of Industry 4.0.

The limitations of this research are related to a single industry type (metal machining) and the influence of single person perspective on the current state of affairs in their company. This can be avoided in the future work, where the individual approach to companies is recommended with several representatives taken from the company employee pool, but also to broaden the research to other industry types and geographical regions. This gives the opportunity to create a specialized audit form regarding the company size and type. Additionally, the presented approach is recommended to be broadened to encompass other departments, in order to finally create a detailed model that enables the management of readiness factor calculation for the entire company and, ultimately, to result in a straightforward definition of an optimal strategic plan for the company in question.

**Author Contributions:** Conceptualization, M.T., H.C. and T.O. methodology, M.T. & T.O.; software, M.T. & N.T.; validation, H.C., T.O. and M.T.; formal analysis, M.T.; investigation, M.T. & N.T.; resources, M.T.; data curation, M.T.; writing—Original draft preparation, M.T.; writing—Review and editing, M.T., & T.O.; visualization, M.T. All authors have read and agreed to the published version of the manuscript.

**Funding:** This research received no external funding.

**Conflicts of Interest:** The authors declare no conflict of interest.

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
