# Peer review of "Process Planning in Industry 4.0—Current State, Potential and Management of Transformation"

_sustainability, doi:10.3390/su12155878_

Round 1

Reviewer 1 Report

Major comments:

  • Row 67-68 (readiness level of 2.15 in Croatia) doesn’t make sense if the readiness model and the reachable scores are not introduced before. It would take too much time to do this in the introduction section. I would suggest to eliminate this side information
  • The definition and explanation of CAPP systems is only marginal although the whole paper is about CAPP – please include a section with literature review about the development of CAPP in the context with Industry 4.0
  • The methodology used for the survey is not described enough. There is missing a profound section about survey design
  • There is no indication what kind of definition is used for the classification of micro, small, medium and large enterprises
  • In Figure 2 you indicate that one group is micro and craft enterprise. Those that are micro see point before. For the ones that are “craft” there is no information or table that shows how many companies are “craft” and what is the classification criteria here
  • Figure 2 shows “high and medium level of education” … there is no information how this data has been measured/collected and how many companies fall in the one or other category
  • Figure 7 is talking about big data manipulation but the question was only about the existence of databases in the companies. The authors did not ask if this data is also used for big data manipulation. As there are many sources in literature that confirm that most companies use only a marginal amount of data for data analytics this is the wrong interpretation of such a question. Authors should correct writing about “Existence of big data in databeses”
  • Figure 11 is not well explained. How many of the companies have this value chain (I guess not all companies have a construction site). This should be indicated. The textual description seems to explain that authors asked participants about the amount of communication with the single players in the value chain. Communication is not horizontal (data) integration. There should be asked if there is a direct data connection/integration (e.g. by using EDI, ERP interfaces,…).
  • The sense of Figure 12 is questionable as it is not clear what was asked in regard to “flexibility” and “change” – the way of asking influences a lot the outcome of such a question. The authors need to describe in a better ways the question.
  • Friedman test should be explained and not only mentioned
  • Table 1 consists of several tables and not only 1

Minor comments:

  • revision of English language is needed
  • CAPP, CAD, CAM needs to be introduced as a term
  • Figure 2 is a screenshot but no reference to the original source is indicated

Author Response

Dear Reviewer 1,

thank you for your comments, I have found them very useful. Please find my response and explanation below, point-by-point and the revised paper attached.

Row 67-68 (readiness level of 2.15 in Croatia) doesn’t make sense if the readiness model and the reachable scores are not introduced before. It would take too much time to do this in the introduction section. I would suggest to eliminate this side information

Thank you for the notice, the information was eliminated from the paper.

The definition and explanation of CAPP systems is only marginal although the whole paper is about CAPP – please include a section with literature review about the development of CAPP in the context with Industry 4.0

The definition and explanation of CAPP systems, along with the introduction of CAD and CAM is now added in chapter 2 "Process planning and Industry 4.0". The recent findings in the scientific literature about the CAPP in the context of industry 4.0 along with its elements which are found to be most challenging at the moment is now described in chapter 2.1.1. CAPP and Industry 4.0 as a literature review. 

The methodology used for the survey is not described enough. There is missing a profound section about survey design

The description of methodology used is added and now gives a more detailed overview about the survey structure, content and grouping of the participants (starting in row 300) 

There is no indication what kind of definition is used for the classification of micro, small, medium and large enterprises

The detailed data about the enterprise size classification by the Croatian law and EU regulation is now added from row 273.

In Figure 2 you indicate that one group is micro and craft enterprise. Those that are micro see point before. For the ones that are “craft” there is no information or table that shows how many companies are “craft” and what is the classification criteria here

The legal definition of craft enterprise by the Croatian law is now added from row 287, while the number of the participants from this group is mentioned in row 324.

Figure 2 shows “high and medium level of education” … there is no information how this data has been measured/collected and how many companies fall in the one or other category

The measurement of educational level is added in row 333 while the number of the participants from certain groups is defined in row 324.

Figure 7 is talking about big data manipulation but the question was only about the existence of databases in the companies. The authors did not ask if this data is also used for big data manipulation. As there are many sources in literature that confirm that most companies use only a marginal amount of data for data analytics this is the wrong interpretation of such a question. Authors should correct writing about “Existence of big data in databases”

Thank you for this very careful and useful notice, Figure 7 is therefore corrected.

Figure 11 is not well explained. How many of the companies have this value chain (I guess not all companies have a construction site). This should be indicated. The textual description seems to explain that authors asked participants about the amount of communication with the single players in the value chain. Communication is not horizontal (data) integration. There should be asked if there is a direct data connection/integration (e.g. by using EDI, ERP interfaces,…).

Concerning the construction site, I think this might be the error in translation. The department the survey was referring to is the product design department, whether it's internal (within the enterprise) or external (outside collaborators). Therefore the change in the formulation of this term has been made (from construction to product design), as well as the question was redefined from the "horizontal integration" to "Communication within the value chain" in chapter 3.6 (row 513). 

The sense of Figure 12 is questionable as it is not clear what was asked in regard to “flexibility” and “change” – the way of asking influences a lot the outcome of such a question. The authors need to describe in a better way the question.

The difference between and importance of human flexibility, frequency of the change and their current subjectivity level in the process is now explained in detail in context of both process planning and industry 4.0 perspective (row 546).

Friedman test should be explained and not only mentioned

The description of the statistical methods used - Friedman test and Mann-Whitney U test and the reason for their use is now added from row 342.

Table 1 consists of several tables and not only 1

Thank you for the notice, this was corrected into Table 1 and 2.

Minor comments:

revision of English language is needed

The revision has now been made by someone fluent in the English language..

CAPP, CAD, CAM needs to be introduced as a term

CAPP, CAD and CAM are now introduced and explained in the context of this research and use in Industry 4.0 from row 106.

Figure 2 is a screenshot but no reference to the original source is indicated

Figure 2 was designed by authors to enhance the quality of research results explanation, and it was based on previous research and literature review given in this paper.

Reviewer 2 Report

The literature and theoretical basis of the paper are very insufficient. Authors should add more good international Journals paper and include it into their analysis.

The paper is lack literature review section. Authors should add it and in this section carefully describe the state-of-the-art in their area of research.

Authors should specify the aim of the paper.

Also, the discussion section is insufficient it lacks the linking with others research from international literature.

It would be good to add limitation of the paper to the conclusion section.

Author Response

Dear Reviewer 2,

thank you for your comments, I have found them very useful. Please find my response and explanation below, point-by-point and the revised paper attached.

The literature and theoretical basis of the paper are very insufficient. Authors should add more good international Journals paper and include it into their analysis.

The paper is lack literature review section. Authors should add it and in this section carefully describe the state-of-the-art in their area of research.

Thank you for the notice, I have added a chapter regarding literature review in the field of Process planning and Industry 4.0. It also deals with the use of CAPP in Industry 4.0 and its most challenging elements based on the recent research results (Chapter 2).

Authors should specify the aim of the paper.

The aim of the paper is now specified more detaily and directly, as a conclusion from the literature review (row 240) and as the aim of research conducted (row 249)

Also, the discussion section is insufficient it lacks the linking with others research from international literature.

This kind of approach to process planning in Industry 4.0 represents a novel approach in such research, so it was roughly compared to the previous research conducted about the Industry 4.0 in manufacturing found in the literature. The regional aspect of the research has been enhanced and the differences between the results are noticed. (row 656).

It would be good to add limitation of the paper to the conclusion section

Thank you for the notice, the limitation is added now to conclusion (row 718), which is now linked to the possibilities of the future work.

Reviewer 3 Report

Dear authors,

this topic is very interesting, and I consider it for really original manuscript.

The paper has scientific soundness, I recommend it to publish after correcting these parts:

Firstly, add to Introduction short Literature Review to the solved issue of Industry 4.0.

Secondly, Table 1, Table 2: use decimal point for all values. Check all manuscript again.

Thirdly, Discussion: focus also on comparing your results to findings of other studies.

I hope my comment will be useful for your future work.

Author Response

Dear Reviewer 3,

thank you for your comments, I have found them very useful. Please find my response and explanation below, point-by-point and revised paper attached.

Firstly, add to Introduction short Literature Review to the solved issue of Industry 4.0.

Thank you for the notice, I have added a chapter regarding literature review in the field of Process planning and Industry 4.0. It also deals with the use of CAPP in Industry 4.0 and its most challenging elements based on the recent research results (Chapter 2).

Secondly, Table 1, Table 2: use decimal point for all values. Check all manuscript again.

Thank you for the notice, the decimal point is now used in the manuscript.

Thirdly, Discussion: focus also on comparing your results to findings of other studies.

This kind of approach to process planning in Industry 4.0 is a novelty research, so it was roughly compared to the previous research conducted about the Industry 4.0 in manufacturing found in the literature. The regional aspect of the research has been enhanced and the differences between the results are noticed. (row 656).

Round 2

Reviewer 1 Report

After reading the first version of the manuscript I was not convinced that authors would be able to achieve a publishable quality of the paper. The authors did a great job in addressing all the suggested points in the revision of the paper. After integrating all comments from different reviewers the quality of the manuscript increased a lot. In my opinion the paper can now be accepted.

Author Response

Dear Reviewer 1, 

thank you for very useful comments.

Reviewer 2 Report

Authors implemented all my remarks.

Author Response

Dear Reviewer 2,

thank you for your very useful comments.

Reviewer 3 Report

Dear authors,

I appreciate your effort and work.

I have only 2 minor recommendations:

Firstly, add to the end of the Introduction aim of the paper and present the structure of the paper.

Secondly, to extend your Introduction or Literature review part I suggest also papers related to the innovation:

Durana, P.; Zauskova, A.; Vagner, L.; Zadnanova, S. Earnings Drivers of Slovak Manufacturers: Efficiency Assessment of Innovation Management. Appl. Sci. 202010, 4251. https://doi.org/10.3390/app10124251

Durana, P.; Valaskova, K.; Vagner, L.; Zadnanova, S.; Podhorska, I.; Siekelova, A. Disclosure of Strategic Managers’ Factotum: Behavioral Incentives of Innovative Business. Int. J. Financial Stud. 20208, 17. https://doi.org/10.3390/ijfs8010017

Good luck.

I hope my comment will be useful for your future work.

Author Response

Dear Reviewer 3,

thank you once again for your very useful comments. I will present you the improvements made point-by-point and please find the revised version of manuscript attached.

Firstly, add to the end of the Introduction aim of the paper and present the structure of the paper.

The aim of the paper and its structure is now added in the Introduction chapter (row 86).

Secondly, to extend your Introduction or Literature review part I suggest also papers related to the innovation:

Durana, P.; Zauskova, A.; Vagner, L.; Zadnanova, S. Earnings Drivers of Slovak Manufacturers: Efficiency Assessment of Innovation Management. Appl. Sci. 2020, 10, 4251. https://doi.org/10.3390/app10124251

Durana, P.; Valaskova, K.; Vagner, L.; Zadnanova, S.; Podhorska, I.; Siekelova, A. Disclosure of Strategic Managers’ Factotum: Behavioral Incentives of Innovative Business. Int. J. Financial Stud. 2020, 8, 17. https://doi.org/10.3390/ijfs8010017

Thank you for pointing out this two very useful papers for my research, I have added them in the Introduction chapter and cited them (row 43, references 4&5)

Thank you once again for very useful comments!

Best regards
